# RaSA: Rank-Sharing Low-Rank Adaptation

**Zhiwei He**[1][*]  **Zhaopeng Tu**[2][†]  **Xing Wang**[2]  **Xingyu Chen**[1][*]  **Zhijie Wang**[1][*]
**Jiahao Xu**[2]  **Tian Liang**[2]  **Wenxiang Jiao**[2]  **Zhuosheng Zhang**[1]  **Rui Wang**[1][†]

[1]Shanghai Jiao Tong University  [2]Tencent AI Lab

[1]{zwhe.cs,galaxychen,violetevergarden,zhangzs,wangrui12}@sjtu.edu.cn
[2]{zptu,brightxwang,jettexu,ttianliang,joelwxjiao}@tencent.com

## Abstract

Low-rank adaptation (LoRA) has been prominently employed for parameter-efficient fine-tuning of large language models (LLMs). However, the limited expressive capacity of LoRA, stemming from the low-rank constraint, has been recognized as a bottleneck, particularly in rigorous tasks like code generation and mathematical reasoning. To address this limitation, we introduce Rank-Sharing Low-Rank Adaptation (RaSA), an innovative extension that enhances the expressive capacity of LoRA by leveraging partial rank sharing across layers. By forming a shared rank pool and applying layer-specific weighting, RaSA effectively increases the number of ranks without augmenting parameter overhead. Our theoretically grounded and empirically validated approach demonstrates that RaSA not only maintains the core advantages of LoRA but also significantly boosts performance in challenging code and math tasks. Code, data and scripts are available at: https://github.com/zwhe99/RaSA.

## 1 Introduction

Low-rank adaptation (LoRA, Hu et al. (2022)) has become a de facto parameter-efficient fine-tuning (PEFT) method for adapting large language models (LLMs) to specific downstream tasks. Its core idea is to constrain the parameter updates to be low-rank, which significantly reduces the number of trainable parameters and allows them to be merged back into the original model, thereby avoiding additional inference latency. Despite its advantages, recent studies have shown that LoRA still lags behind full fine-tuning (FFT), particularly in scenarios involving large training datasets and complex tasks such as mathematical reasoning and code generation (Jiang et al., 2024; Biderman et al., 2024). A plausible explanation for this performance gap is that the low-rank constraint limits the expressive capacity of LoRA. For instance, Biderman et al. (2024) empirically found that the effective rank required for FFT is 10-100× higher than typical LoRA configuration, and Zeng & Lee (2024) theoretically demonstrated that a Transformer network (Vaswani et al., 2017) requires a rank at least half the size of the model dimension to approximate another model of similar size.

Although the limited number of trainable parameters results in limited expressive capacity, recent studies still indicate redundancy in LoRA's parameters. For example, Kopiczko et al. (2024); Song et al. (2024); Renduchintala et al. (2024); Li et al. (2024) further reduced the number of LoRA's parameters by sharing them across layers and modules with only slight performance loss. Brüel-Gabrielsson et al. (2024) compressed 1,000 LoRAs trained from different tasks by sharing their parameter spaces. This contradiction suggests that LoRA's parameters are still not being fully utilized.

Combining the above two observations, we propose **Ra**nk-**S**haring Low-Rank **A**daptation (RaSA), an approach that boosts the expressive capacity of LoRA by enabling partial rank sharing across layers. Specifically, given an LLM with $L$ layers, RaSA extracts $k$ ranks from each layer's LoRA update to form a rank pool of $L \times k$ ranks, which is shared across all layers with layer-specific weighting. RaSA retains the core advantages of LoRA – keeping the same parameter overhead and allowing for easy merging back into the model. Moreover, since modern LLMs typically have deep architectures (i.e., large $L$), RaSA greatly increase the effective rank of the parameter update by $(L - 1) \times k$.

---

[*]Work was done when Zhiwei He, Xingyu Chen, and Zhijie Wang were interning at Tencent AI Lab.
[†]Zhaopeng Tu and Rui Wang are co-corresponding authors.

However, a higher rank does not necessarily lead to better expressive capacity. To rigorously assess the benefits of RaSA, we analyze its capacity to reconstruct high-rank matrices compared to LoRA. Theoretically, we prove that RaSA's minimum reconstruction error is bounded by that of LoRA. Empirically, we show that when $k$ is relatively small, RaSA can be easily optimized to achieve a significantly lower reconstruction error than LoRA. Finally, we conducted experiments on mathematical reasoning and code generation, demonstrating that the lower reconstruction error translates to improved downstream task performance.

Our contributions are summarized as follows:

- We propose RaSA, a novel extension of LoRA by by allowing partial rank sharing across layers, which significantly improves the method's efficiency and expressiveness (§ 2).
- We provide a comprehensive analysis – both theoretical and empirical – showcasing RaSA's superior capacity for matrix reconstruction (§ 3) and its resultant improved performance on rigorous downstream tasks (e.g. code and math) (§ 4).

## 2 METHOD

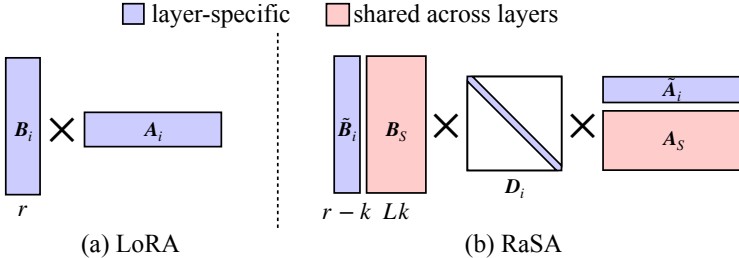

Figure 1: Decomposition of the update matrix $\Delta \boldsymbol{W}_i$ in LoRA and RaSA, where $i$ is the layer index.

### 2.1 FORMULATION

Given a pre-trained weight matrix $\boldsymbol{W} \in \mathbb{R}^{b \times a}$, LoRA constrains its update to a low-rank form by decomposing the update matrix $\Delta \boldsymbol{W} \in \mathbb{R}^{b \times a}$ into a product of two rank-$r$ matrices:

$$\boldsymbol{W} + \Delta \boldsymbol{W} = \boldsymbol{W} + \frac{\alpha}{r} \boldsymbol{B} \boldsymbol{A} \quad (\boldsymbol{B} \in \mathbb{R}^{b \times r}, \boldsymbol{A} \in \mathbb{R}^{r \times a}), \tag{1}$$

where rank $r \ll \min(b, a)$ serves as a bottleneck dimension, reducing the number of trainable parameters, and $\alpha$ is a scaling factor. In an LLM with $L$ layers, LoRA assigns distinct trainable matrices to each layer-$i$: $\{\boldsymbol{B}_i \boldsymbol{A}_i\}_{i \in [L]}$ (Figure 1(a)). RaSA, on the other hand, mitigates the low-rank bottleneck of LoRA through rank sharing. Specifically, RaSA takes out $k$ ranks in each layer and shares them across all layers. This process can be conceptualized as follows:

1. Split the matrices $\boldsymbol{B}_i$ and $\boldsymbol{A}_i$ into layer-specific parts ($\tilde{\boldsymbol{B}}_i$, $\tilde{\boldsymbol{A}}_i$) and layer-shared parts ($\hat{\boldsymbol{B}}_i$, $\hat{\boldsymbol{A}}_i$):

$$\boldsymbol{B}_i = [\ \underbrace{\tilde{\boldsymbol{B}}_i}_{\mathbb{R}^{b \times (r-k)}} \quad \underbrace{\hat{\boldsymbol{B}}_i}_{\mathbb{R}^{b \times k}}], \quad \boldsymbol{A}_i = [\ \underbrace{\tilde{\boldsymbol{A}}_i^T}_{\mathbb{R}^{a \times (r-k)}} \quad \underbrace{\hat{\boldsymbol{A}}_i^T}_{\mathbb{R}^{a \times k}}]^T. \tag{2}$$

2. Concatenate the layer-shared parts across all layers to form shared rank pools ($\boldsymbol{B}_S$ and $\boldsymbol{A}_S$):

$$\boldsymbol{B}_S = \begin{bmatrix} \hat{\boldsymbol{B}}_1 & \hat{\boldsymbol{B}}_2 & \cdots & \hat{\boldsymbol{B}}_L \end{bmatrix} \in \mathbb{R}^{b \times (L \times k)}, \quad \boldsymbol{A}_S = \begin{bmatrix} \hat{\boldsymbol{A}}_1^T & \hat{\boldsymbol{A}}_2^T & \cdots & \hat{\boldsymbol{A}}_L^T \end{bmatrix}^T \in \mathbb{R}^{(L \times k) \times a}. \tag{3}$$

Therefore, the update for layer-$i$ is given by:

$$\begin{aligned} \boldsymbol{W}_i + \Delta \boldsymbol{W}_i &= \boldsymbol{W}_i + \frac{\alpha}{r} (\tilde{\boldsymbol{B}}_i \tilde{\boldsymbol{A}}_i + \boldsymbol{B}_S \boldsymbol{A}_S) \\ &= \boldsymbol{W}_i + \begin{bmatrix} \tilde{\boldsymbol{B}}_i & \boldsymbol{B}_S \end{bmatrix} \operatorname{diag}(\frac{\alpha}{r}) \begin{bmatrix} \tilde{\boldsymbol{A}}_i \\ \boldsymbol{A}_S \end{bmatrix}. \end{aligned} \tag{4}$$

To enable layer-specific weighting, we replace the constant diagonal matrix with a trainable diagonal matrix $\boldsymbol{D}_i = \text{diag}(d_1, d_2, \cdots, d_j, \cdots, d_{r-k+Lk})$, yielding the final RaSA update (Figure 1(b)):

$$\boldsymbol{W}_i + \Delta\boldsymbol{W}_i = \boldsymbol{W}_i + \underbrace{\begin{bmatrix} \tilde{\boldsymbol{B}}_i & \boldsymbol{B}_S \end{bmatrix}}_{\mathbb{R}^{b\times(r-k+Lk)}} \boldsymbol{D}_i \underbrace{\begin{bmatrix} \tilde{\boldsymbol{A}}_i \\ \boldsymbol{A}_S \end{bmatrix}}_{\mathbb{R}^{(r-k+Lk)\times a}}. \tag{5}$$

## 2.2 ANALYSIS & IMPLEMENTATION DETAILS

**Rank of $\Delta\boldsymbol{W}$** Comparing Equations (1) and (5), RaSA increases the rank of $\Delta\boldsymbol{W}$ from $r$ to $r - k + Lk$. Since modern LLMs are deep, RaSA significantly boosts the model's expressive capacity by enabling a higher effective rank, on which we have a detailed discussion in § 3. Each layer in RaSA maintains the same rank for $\Delta\boldsymbol{W}$, which sets it apart from methods that dynamically assign ranks across layers, such as AdaLoRA (Zhang et al., 2023) and PriLoRA (Benedek & Wolf, 2024).

**Additional Parameters** RaSA introduces the diagonal matrix $\boldsymbol{D}_i$ as additional parameters. Since $\boldsymbol{D}_i$ is diagonal and operates only at the bottleneck dimension, the added parameters are negligible. In practice, $\boldsymbol{D}_i$ contributes to less than 0.001% of the total model parameters.

**Initialization** Following LoRA, we use Kaiming initialization (He et al., 2015) for $\tilde{\boldsymbol{A}}_i$ and $\boldsymbol{A}_S$, and initialize $\tilde{\boldsymbol{B}}_i$ and $\boldsymbol{B}_S$ to zero. For $\boldsymbol{D}_i$, we differentiate between the layer-specific and layer-shared parts by scaling $\alpha$ proportionally by their respective ranks:

$$d_j = \begin{cases} \frac{1}{2} \times \frac{\alpha}{r-k} & \text{if } j \leq r - k, \\ \frac{1}{2} \times \frac{\alpha}{Lk} & \text{if } j > r - k. \end{cases} \tag{6}$$

**Same Dimension Assumption** RaSA assumes that all layers share the same dimensionality. This holds for the vast majority of models (e.g. Llama (Dubey et al., 2024), Mistral (Jiang et al., 2023)).

## 3 RECONSTRUCTION ERROR ANALYSIS

While RaSA increases the effective rank of $\Delta\boldsymbol{W}$, a higher rank does not necessarily guarantee improved expressive capacity. For instance, a full-rank identity matrix can only perform the identity transformation. To assess the expressive capacity of LoRA and RaSA, we compare their abilities to reconstruct a set of high-rank matrices $\{\boldsymbol{M}_i\}_{i\in[L]}$, where $\text{rank}(\boldsymbol{M}_i) = R > r$. Under the Frobenius norm, the **minimum reconstruction error (MRE) of LoRA** is defined as:

$$e_{\text{lora}} = \min_{\boldsymbol{B}_i, \boldsymbol{A}_i} \sum_{i=1}^{L} \|\boldsymbol{M}_i - \boldsymbol{B}_i\boldsymbol{A}_i\|_F^2. \tag{7}$$

According to the Eckart–Young–Mirsky theorem (Eckart & Young, 1936), we can perform singular value decomposition (SVD) on $\boldsymbol{M}_i$:

$$\text{SVD}(\boldsymbol{M}_i) = \sum_{j=1}^{R} \sigma_j^{(i)} \boldsymbol{u}_j^{(i)} \boldsymbol{v}_j^{(i)T} (\sigma_1^{(i)} \geq \sigma_2^{(i)} \geq \cdots \geq \sigma_R^{(i)}). \tag{8}$$

LoRA's optimal approximation is given by the first $r$ components of $\text{SVD}(\boldsymbol{M}_i)$, and $e_{\text{lora}}$ becomes the sum of squares of the discarded singular values (those beyond the $r$-th one):

$$e_{\text{lora}} = \sum_{i=1}^{L} \|\boldsymbol{M}_i - \sum_{j=1}^{r} \sigma_j^{(i)} \boldsymbol{u}_j^{(i)} \boldsymbol{v}_j^{(i)T}\|_F^2 = \sum_{i=1}^{L} \sum_{j=r+1}^{R} \sigma_j^{(i)2}. \tag{9}$$

Similarly, when each layer shares $k$ ranks out, we can define the **MRE of RaSA** as:

$$e_{\text{rasa}(k)} = \min_{\tilde{\boldsymbol{B}}_i, \tilde{\boldsymbol{A}}_i, \boldsymbol{B}_S, \boldsymbol{A}_S, \boldsymbol{D}_i} \sum_{i=1}^{L} \|\boldsymbol{M}_i - \begin{bmatrix} \tilde{\boldsymbol{B}}_i & \boldsymbol{B}_S \end{bmatrix} \boldsymbol{D}_i \begin{bmatrix} \tilde{\boldsymbol{A}}_i \\ \boldsymbol{A}_S \end{bmatrix}\|_F^2. \tag{10}$$

For simplicity, in this section we consider that $\boldsymbol{D}_i$ operates only on the shared matrices $\boldsymbol{B}_S$ and $\boldsymbol{A}_S$, which does not affect the value of $e_{\text{rasa}(k)}$:

$$e_{\text{rasa}(k)} = \min_{\tilde{\boldsymbol{B}}_i, \tilde{\boldsymbol{A}}_i, \boldsymbol{B}_S, \boldsymbol{A}_S, \boldsymbol{D}_i} \sum_{i=1}^{L} \|\boldsymbol{M}_i - (\tilde{\boldsymbol{B}}_i\tilde{\boldsymbol{A}}_i + \boldsymbol{B}_S\boldsymbol{D}_i\boldsymbol{A}_S)\|_F^2. \tag{11}$$

### 3.1 THEORETICAL ANALYSIS

**Theorem 3.1.** $e_{\text{rasa}(k)} \leq e_{\text{lora}}$

*Proof.* To prove this, we construct a feasible solution for RaSA that achieves the same reconstruction error as LoRA's minimum error. This is done by distributing the ranks shared across layers in RaSA such that they cover the same rank range as the optimal LoRA solution.

For each layer-$i$, we take the last $k$ components (corresponding to the indices $r - k + 1$ through $r$) from the LoRA's optimal approximation (Equation (9)), forming the following matrices:

$$\boldsymbol{U}^{(i)} = \begin{bmatrix} \boldsymbol{u}_{r-k+1}^{(i)} & \boldsymbol{u}_{r-k+2}^{(i)} & \cdots & \boldsymbol{u}_r^{(i)} \end{bmatrix},$$
$$\boldsymbol{V}^{(i)} = \begin{bmatrix} \boldsymbol{v}_{r-k+1}^{(i)} & \boldsymbol{v}_{r-k+2}^{(i)} & \cdots & \boldsymbol{v}_r^{(i)} \end{bmatrix}, \tag{12}$$
$$\boldsymbol{\Sigma}^{(i)} = \begin{bmatrix} \sigma_{r-k+1}^{(i)} & \sigma_{r-k+2}^{(i)} & \cdots & \sigma_r^{(i)} \end{bmatrix}.$$

The shared matrices $\boldsymbol{B}_S$ and $\boldsymbol{A}_S$ are constructed by stacking $\boldsymbol{U}^{(i)}$ and $\boldsymbol{V}^{(i)}$ from each layer:

$$\boldsymbol{B}_S = \begin{bmatrix} \boldsymbol{U}^{(1)} & \cdots & \boldsymbol{U}^{(i)} & \cdots & \boldsymbol{U}^{(L)} \end{bmatrix}, \quad \boldsymbol{A}_S = \begin{bmatrix} \boldsymbol{V}^{(1)} & \cdots & \boldsymbol{V}^{(i)} & \cdots & \boldsymbol{V}^{(L)} \end{bmatrix}^T. \tag{13}$$

Similarly, we define the diagonal matrix $\boldsymbol{D}_i$ for each layer-$i$ by placing the corresponding singular values $\boldsymbol{\Sigma}^{(i)}$ in their appropriate positions:

$$\boldsymbol{D}_i = \text{diag}(\begin{bmatrix} \boldsymbol{0} & \cdots & \boldsymbol{\Sigma}^{(i)} & \cdots & \boldsymbol{0} \end{bmatrix}). \tag{14}$$

Finally, the matrices $\tilde{\boldsymbol{B}}_i$ and $\tilde{\boldsymbol{A}}_i$ are formed from the first $r - k$ components of $\text{SVD}(\boldsymbol{M}_i)$:

$$\tilde{\boldsymbol{B}}_i \tilde{\boldsymbol{A}}_i = \sum_{j=1}^{r-k} \sigma_j^{(i)} \boldsymbol{u}_j^{(i)} \boldsymbol{v}_j^{(i)T}. \tag{15}$$

Substituting Equations (13) to (15) into Equation (11), we derive the following:

$$
\begin{aligned}
e_{\text{rasa}(k)} &\leq \sum_i^L \| \boldsymbol{M}_i - (\tilde{\boldsymbol{B}}_i \tilde{\boldsymbol{A}}_i + \boldsymbol{B}_S \boldsymbol{D}_i \boldsymbol{A}_S) \|_F^2 \\
&= \sum_{i=1}^L \| \sum_{j=1}^R \sigma_j^{(i)} \boldsymbol{u}_j^{(i)} \boldsymbol{v}_j^{(i)T} - (\sum_{j=1}^{r-k} \sigma_j^{(i)} \boldsymbol{u}_j^{(i)} \boldsymbol{v}_j^{(i)T} + \sum_{j=r-k+1}^r \sigma_j^{(i)} \boldsymbol{u}_j^{(i)} \boldsymbol{v}_j^{(i)T}) \|_F^2 \\
&= \sum_{i=1}^L \| \sum_{j=1}^R \sigma_j^{(i)} \boldsymbol{u}_j^{(i)} \boldsymbol{v}_j^{(i)T} - \sum_{j=1}^r \sigma_j^{(i)} \boldsymbol{u}_j^{(i)} \boldsymbol{v}_j^{(i)T} \|_F^2 \\
&= \sum_{i=1}^L \sum_{j=r+1}^R \sigma_j^{(i)2} \\
&= e_{\text{lora}}.
\end{aligned}
\tag{16}
$$

Thus, we conclude that $e_{\text{rasa}(k)} \leq e_{\text{lora}}$, proving that RaSA can achieve equal or lower minimum reconstruction error compared to LoRA. $\square$

### 3.2 EMPIRICAL ANALYSIS

While the previous theoretical analysis guarantees that RaSA can at least match the MRE of LoRA, it does not quantify how much RaSA improves upon LoRA. To provide a more intuitive understanding of how RaSA achieves lower reconstruction error, we turn to an optimization-based analysis using coordinate descent.

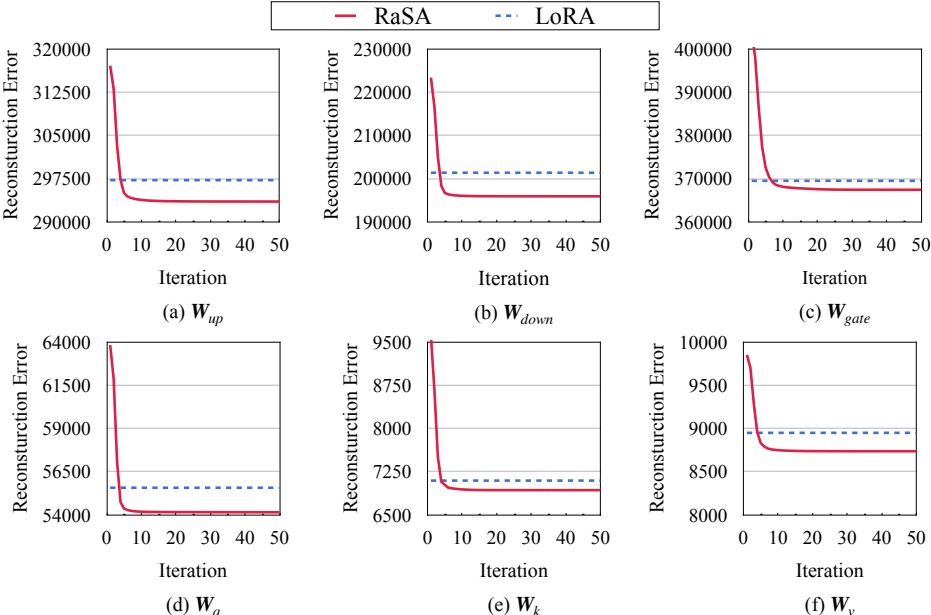

Figure 2: Reconstruction error curves of RaSA ($r = 8, k = 1$) during coordinate descent. We also plot the minimum reconstruction error of LoRA (Equation (9)) for comparison.

**Empirical Validation** Specifically, we instantiate the set of high-rank matrices $\{M_i\}_{i \in [L]}$ with the actual weight updates from model fine-tuning: $\{\Delta W_i\}_{i \in [L]}$, and iteratively minimize the reconstruction error in Equation (11) by adjusting the parameters of RaSA ($r = 8, k = 1$), namely the $\tilde{B}_i$, $\tilde{A}_i$, $B_S$, $A_S$, and $D_i$ (details can be found in appendix A). We apply this procedure to various kinds of linear modules within Llama-3.1-8B until convergence, and compute $e_{\text{lora}}$ using Equation (9) as baseline values.

Figure 2 shows that RaSA requires ∼10 iterations to achieve a significantly lower reconstruction error than LoRA's minimum. This pattern is consistent across all linear modules in the model, demonstrating the enhanced expressive capacity of RaSA.

**Selection of $k$** RaSA introduces only one additional hyper-parameter, $k$, which controls how many ranks are taken from each layer to be shared across all layers. When $k = 0$, RaSA reduces to LoRA, where no ranks are shared. On the other hand, when $k = r$, RaSA shares all ranks across layers, eliminating layer-specific low-rank updates and making the adaptation fully shared. While this maximizes the effective rank of update, it may diminish layer diversity and the ability to capture layer-specific nuances. We traversed $k$ from 0 to 8, and presents the converged reconstruction error from the previous coordinate descent experiment in Figure 3. The results indicate that a small value of $k$, around $r/8$, achieves the minimum error. Further increasing $k$ can lead to a rise in reconstruction error, even exceeding that of LoRA. This finding also indicates that some current methods that share all ranks across all layers, such as VeRA (Kopiczko et al., 2024) and Tied-LoRA (Renduchintala et al., 2024), might be sub-optimal and challenging to be optimized.

# 4 EXPERIMENT

## 4.1 SETUP

**Tasks** Our experiments generally align with those reported by Biderman et al. (2024). We applied all the methods to instruction fine-tuning and evaluated their performance on challenging tasks: code generation and mathematical reasoning. While Biderman et al. (2024) use Humaneval (Chen et al., 2021) and GSM8K (Cobbe et al., 2021) as test sets, these two benchmarks have become saturated with the rapid growth of LLMs. To provide a more rigorous evaluation, we adopted two more challenging

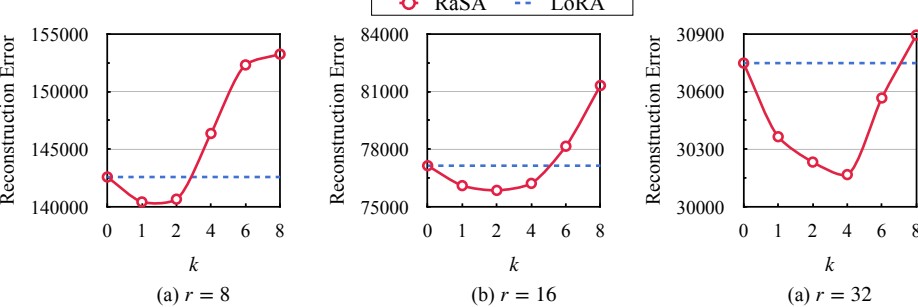

Figure 3: Reconstruction error comparison between RaSA and LoRA as a function of the shared rank parameter $k$. We also plot the minimum reconstruction error of LoRA (Equation (9)) for comparison. The results are average across all linear modules in the model.

benchmarks as test sets. Since these two benchmarks lack validation sets, in addition to reporting the results from the last checkpoint, we also report the best results as a reference for the upper bound of each method. Prompt templates for evaluation are provided in appendix B.

- *Code Generation*: We used Magicoder-Evol-Instruct-110k (Wei et al., 2024) as the training data, a collection of programming question-answer (QA) pairs, which is a reproduced and decontaminated version of WizardCoder (Luo et al., 2024). We used Humaneval+ (Liu et al., 2023) as the test set, an extension of the Humaneval benchmark that scales the number of test cases by $80\times$. We used the Bigcode Evaluation Harness (Ben Allal et al., 2022) as the evaluation tool, sampling 50 solutions per problem with a temperature of 0.2, and report both Pass@1 and Pass@10.
- *Mathematical Reasoning*: We used MetaMathQA (Yu et al., 2024) as the training data, which comprises 395K QA pairs derived from the training sets of GSM8K (Cobbe et al., 2021) and MATH (Hendrycks et al., 2021), rewritten by GPT-3.5. We used MATH (Hendrycks et al., 2021) as the test set, which consists of 5K competition-level mathematics problems covering 7 subjects and 5 difficulty levels. We followed the evaluation protocol from LLMs Evaluation Harness (Gao et al., 2024), using sympy to verify correctness and employing greedy search for generation.

**Baselines**    We compare RaSA to the several representative PEFT methods:

- LoRA (Hu et al., 2022) that learns only a low-rank perturbation to the pretrained weight matrix.
- MoRA (Jiang et al., 2024) that uses block diagonal matrices instead of low-rank matrices.
- VeRA (Kopiczko et al., 2024) that fully shares the low-rank matrices across all layers with layer-specific weighting, and freeze the low-rank matrices during training to achieve extreme parameter efficient fine-tuning. Therefore, VeRA can set a higher rank-$r$ than LoRA.

**LLMs & Training Details**    We conducted experiments on two open-sourced LLMs: Llama-3.1-8B (Dubey et al., 2024) and Mistral-0.3-7B (Jiang et al., 2023). Following common practice (Kopiczko et al., 2024; Jiang et al., 2024), we used pre-trained models rather than instruction-tuned ones. We applied PEFTs on all linear modules from attention ($W_q, W_k, W_v, W_o$) and feed-forward networks ($W_{up}, W_{down}, W_{gate}$). We set the model hyper-parameters based on the optimal configurations from Biderman et al. (2024), employing the decoupled LionW optimizer with a batch size of 192, and training for 8 epochs with a learning rate of 5e-4 by default. For RaSA, we set $k = \max(r/8, 1)$ based on the analysis in § 3.2. More details are provided in appendix C.

## 4.2    MAIN RESULTS

In this section, we compare RaSA and baselines on two challenging domains – code and math.

**Code Generation**    Table 1 presents the results on the Humaneval+ test set. We compare RaSA and prior LoRA variants in terms of both efficiency and effectiveness. Although VeRA adds only 1.6M extra parameters for $r = 1024$, it results in a training time increase of between 13% and 16% over

Table 1: Performance on the code generation task (i.e. Humaneval+). We *italicize* the best result for each rank, and **bold** the best result for each model. We also present training cost for each setting in terms of trainable parameters (**# Trainable Param.**) and training time (**Time**). Note that for MoRA and RaSA, $r$ does not correspond to the effective rank of the update matrix.

| r | Method | # Trainable Param. | Llama-3.1-8B | | | | | Mistral-0.3-7B | | | | |
|---|---|---|---|---|---|---|---|---|---|---|---|---|
| | | | Time | PASS@1 | | PASS@10 | | Time | PASS@1 | | PASS@10 | |
| | | | | BEST | LAST | BEST | LAST | | BEST | LAST | BEST | LAST |
| 1024 | VeRA | 1.6M | 11.3h | 48.8 | 48.8 | 66.5 | 64.2 | 12.5h | 42.5 | 39.5 | 57.3 | 54.4 |
| 8 | LoRA | 21.0M | 9.6h | 56.1 | 53.0 | 71.2 | 68.5 | 10.7h | 42.6 | 39.7 | 57.7 | 54.8 |
| | MoRA | 21.0M | 12.0h | 54.6 | 52.1 | 68.4 | 66.9 | 13.4h | 45.2 | 38.6 | 64.4 | 48.6 |
| | RaSA | 21.0M | 11.2h | *57.9* | **56.9** | **72.6** | *69.6* | 12.1h | *50.0* | *49.0* | *66.0* | *64.2* |
| 16 | LoRA | 41.9M | 9.8h | 54.5 | 53.4 | 68.9 | 67.6 | 10.7h | 46.0 | 40.6 | 61.2 | 54.9 |
| | MoRA | 41.9M | 12.7h | 56.3 | 52.9 | 69.5 | 65.6 | 14.0h | 43.4 | 41.0 | 59.4 | 56.0 |
| | RaSA | 42.0M | 11.2h | *57.3* | *56.4* | *72.1* | *68.1* | 12.1h | *53.6* | *51.3* | *68.5* | *63.7* |
| 32 | LoRA | 83.9M | 10.0h | 57.9 | **56.9** | 69.8 | 69.2 | 10.8h | 50.2 | 44.4 | 64.4 | 57.0 |
| | MoRA | 83.9M | 12.4h | 55.6 | 53.0 | 69.0 | 68.3 | 14.0h | 42.2 | 42.2 | 56.4 | 56.0 |
| | RaSA | 83.9M | 11.5h | **59.5** | 56.2 | *72.5* | **71.4** | 12.5h | **55.7** | **55.7** | **70.0** | **65.7** |

LoRA with $r = 32$. VeRA, however, is the least effective among all the variants due to its extreme strategy for parameter efficiency. Both MoRA and RaSA add a comparable number of additional parameters as LoRA, yet MoRA requires more time due to the use of block diagonal matrices.

In terms of model performance, MoRA shows performance on par with LoRA, aligning with the findings reported in the original paper (Jiang et al., 2024). Our proposed RaSA surpasses all baseline models in nearly all scenarios. Like LoRA, RaSA's performance improves with rank, and at rank 32, RaSA typically delivers the strongest performance for both the Llama and Mistral models. RaSA achieves maximum Humaneval+ of 59.5% PASS@1 with Llama-3.1-8B.

Table 2: Performance on mathematical reasoning task (i.e. MATH). We also present extra parameters (**# Extra Param.**) used by AdaLoRA and PriLoRA for estimating parameter importance.

| r | Method | # Trainable Param. | # Extra Param. | Llama-3.1-8B | | | Mistral-0.3-7B | | |
|---|---|---|---|---|---|---|---|---|---|
| | | | | Time | ACC | | Time | ACC | |
| | | | | | BEST | LAST | | BEST | LAST |
| – | FFT | 7-8B | | | 35.5 | 34.6 | | 28.1 | 26.6 |
| 1024 | VeRA | 1.6M | — | 22.4h | 27.4 | 25.6 | 17.6h | 19.9 | 19.4 |
| 8 | LoRA | 21.0M | — | 20.1h | 28.3 | 26.7 | 14.6h | 20.1 | 19.2 |
| | MoRA | 21.0M | — | 23.6h | 29.2 | 28.9 | 19.6h | 21.4 | 21.4 |
| | OLoRA | 21.0M | — | 29.3h | 28.4 | 27.8 | 31.1h | 22.5 | 22.5 |
| | AdaLoRA | 31.5M | 63.0M | 27.9h | *30.4* | 28.9 | 17.7h | 22.5 | 21.6 |
| | PriLoRA | 21.3M | 10.7M | 24.1h | 28.2 | *29.5* | 15.9h | 22.3 | 22.3 |
| | RaSA | 21.0M | — | 23.8h | 30.3 | 29.1 | 15.9h | *24.3* | *23.8* |
| 16 | LoRA | 41.9M | — | 20.2h | 28.8 | 27.1 | 14.7h | 20.9 | 19.5 |
| | MoRA | 41.9M | — | 24.5h | 30.2 | 26.5 | 21.1h | 20.5 | 19.4 |
| | OLoRA | 41.9M | — | 29.6h | 28.6 | 28.4 | 35.9h | 22.5 | 22.2 |
| | AdaLoRA | 62.9M | 125.8M | 28.1h | 29.7 | 29.4 | 17.6h | 23.5 | 23.2 |
| | PriLoRA | 42.6M | 21.3M | 24.5h | 28.6 | 29.4 | 15.9h | 22.7 | 21.6 |
| | RaSA | 42.0M | — | 24.4h | *31.4* | 29.8 | 15.8h | *25.9* | **25.1** |
| 32 | LoRA | 83.9M | — | 20.6h | 28.9 | 27.2 | 14.8h | 21.8 | 20.4 |
| | MoRA | 83.9M | — | 24.7h | 28.6 | 25.8 | 20.5h | 18.4 | 18.4 |
| | OLoRA | 83.9M | — | 29.9h | 29.0 | 28.7 | 31.3h | 23.8 | 23.5 |
| | AdaLoRA | 125.9M | 251.8M | 28.7h | 30.2 | 29.4 | 17.9h | 23.5 | 23.5 |
| | PriLoRA | 85.2M | 42.6M | 24.5h | 30.0 | 28.8 | 16.4h | 24.8 | 24.2 |
| | RaSA | 83.9M | — | 24.3h | *31.7* | **29.6** | 16.5h | **26.1** | **25.1** |

**Mathematical Reasoning**   We add FFT and more PEFT baselines in math task:

- OLoRA (Büyükakyüz, 2024) that uses QR decomposition to initialize the LoRA adapters.

- AdaLoRA (Zhang et al., 2023) that dynamically allocates ranks among parameter matrices.

- PriLoRA (Benedek & Wolf, 2024) that allocates a different rank for each layer in an increasing manner, and performs pruning throughout the training process.

The math results are presented in Table 2. FFT outperforms all PEFT methods, aligning the findings from Biderman et al. (2024). Considering both training cost and accuracy, RaSA demonstrates consistent superiority over all PEFT baselines across various configurations. Mistral notably falls short of its Llama counterpart, exhibiting a performance deficit of approximately 8%, which RaSA is capable of narrowing down to 5%. We also observe that directly increasing the hyper-parameter $r$ yields only marginal performance gains, but at the cost of doubling the number of training parameters. In contrast, RaSA greatly outperforms LoRA with the same or even fewer parameters ($\text{RaSA}_{r=8} > \text{LoRA}_{r=32}$). This supports the notion introduced in § 1 that LoRA's parameters are underutilized. RaSA, on the other hand, improves the utilization of parameters by sharing them across layers.

## 4.3 RaSA Learns More and Forgets Less than LoRA

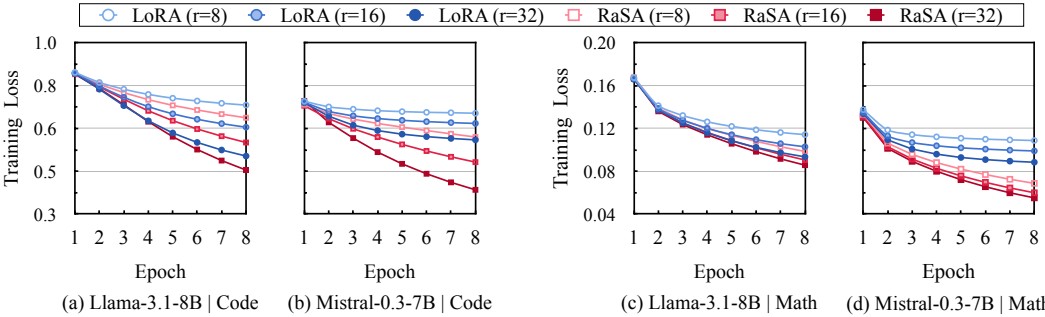

(a) Llama-3.1-8B | Code  (b) Mistral-0.3-7B | Code  (c) Llama-3.1-8B | Math  (d) Mistral-0.3-7B | Math

Figure 4: **RaSA learns more and faster than LoRA**. Training curves of LoRA and RaSA with different ranks. RaSA consistently outperforms LoRA with the same rank across models and tasks.

**RaSA learns more and faster than LoRA**  Figure 4 illustrates the training curves of the fine-tuning process. Generally, the training losses for both RaSA and LoRA decrease as the rank increases. Notably, RaSA consistently outperforms its LoRA counterpart in terms of both learning effectiveness and efficiency across all cases, aligning with our empirical analysis presented in Section 3.2. These results collectively underscore the efficacy and universal applicability of the proposed RaSA method. One interesting finding is that RaSA is specifically effective for `Mistral`: RaSA achieves comparable or potentially superior training outcomes to LoRA with a significantly lower rank requirement of 8, compared to LoRA's rank of 32.

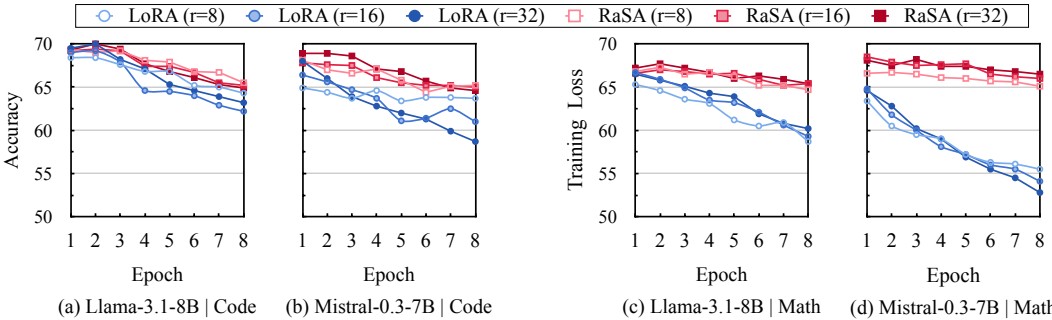

(a) Llama-3.1-8B | Code  (b) Mistral-0.3-7B | Code  (c) Llama-3.1-8B | Math  (d) Mistral-0.3-7B | Math

Figure 5: **RaSA forgets less than LoRA**. Y-axis shows the average of prediction accuracy on three benchmarks to evaluate model's forgetting. Higher prediction accuracy denotes less forgetting.

**RaSA forgets less than LoRA** We follow Biderman et al. (2024) to investigate the extent of forgetting as degradation of base model capabilities. Specifically, we calculate prediction accuracies on the following three benchmarks: (1) HellaSwag (Zellers et al., 2019): inference the most plausible continuation for daily events (70K problems); (2) WinoGrande (Sakaguchi et al., 2019): assesses commonsense reasoning (44K problems); (3) ARC-Challenge (Clark et al., 2018): complex reasoning and understanding of scientific concepts (7.8K problems).

Figure 5 presents the averaged forgetting curves of the three benchmarks, clearly showing that RaSA experiences less forgetting than LoRA, with RaSA's forgetting levels being less affected by rank changes compared to LoRA. The difference in performance between $r = 8$ and $r = 32$ at epoch 8 stands at an average of 2.83% for LoRA and 0.75% for RaSA, indicating a smaller performance variation for RaSA. LoRA is more prone to forgetting in math than code, while RaSA displays greater domain robustness. Specifically, with $r = 32$, Mistral scores 58.7% in code and 52.8% in math using LoRA, whereas RaSA shows a reduced performance difference between code (64.6%) and math (66.5%) domains, underscoring RaSA's robustness.

## 4.4 SCALING PERFORMANCE ANALYSIS

This section investigates the scaling characteristics of the RaSA approach by varying both the model size and the dataset size to assess its robustness.

**Model Scaling** Initially, we evaluate RaSA's performance on an expanded scale by examining larger-scale models, including Llama-3.1-70B and Mixtral-8×7B. Due to computational constraints, we employ a rank of $r = 4$ for both models specifically in the domain of mathematical reasoning. For an equitable comparison, we present results for smaller models configured with $r = 4$. Each model is trained over 2 epochs using both LoRA and RaSA techniques, with performance measured in terms of LAST accuracy. The results in Figure 6 reveal that increasing the model size substantially enhances performance for both LoRA and RaSA, across all model types. Noteworthy is the performance of larger Llama and Mistral models using LoRA, achieving MATH accuracies of 40.4% and 32.6%, respectively. These results significantly exceed those of their smaller counterparts under identical configurations and even surpass outcomes from variants with extended training (i.e., 8 epochs). Notably, RaSA consistently outperforms LoRA on these larger-scale models, underscoring RaSA's robustness in handling models of increased scale.

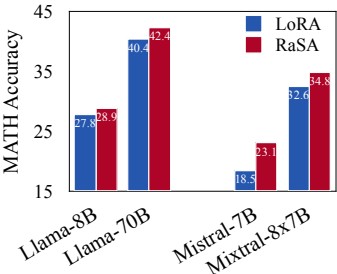

Figure 6: MATH performance of scaled models.

**Data Scaling** Subsequently, we explore the influence of training data size on RaSA's performance. We experiment with the Llama-3.1-8B model, applying a rank of $r = 8$ to facilitate efficient training. The examination involves random sampling of 25% and 50% instances from the SFT data for the mathematics reasoning task. Each model is trained over 8 epochs, with performance assessed through the LAST accuracy. As illustrated in Figure 7, LoRA's performance seems contingent on the volume of training data, with no noticeable improvement when data is increased from 25% to 50%. This finding is consistent with the results in Biderman et al. (2024). In contrast, RaSA demonstrates a remarkable ability to enhance performance with an increase in training data volume. Impressively, with just 25% of the training data, RaSA outperforms LaSA even when the latter utilizes the entire dataset, highlighting RaSA's exceptional efficiency in leveraging training data for performance improvement.

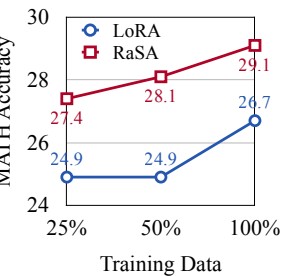

Figure 7: MATH performance of scaled data.

## 5 RELATED WORK

**Parameter-Efficient Fine-Tuning (PEFT)** PEFT methods aim to minimize the number of trainable parameters needed for fine-tuning large models, thus reducing memory and computational require-

ments. Pioneering methods include adapter-based (Houlsby et al., 2019) and prompt-based (Lester et al., 2021; Li & Liang, 2021) that introduce additional tunable adapter or prefix tokens to enable efficient fine-tuning while keeping the original model parameters fixed. However, these approaches can slow down inference speed due to the extra components introduced. LoRA overcomes this drawback by introducing low-rank matrices directly into the weight update process during fine-tuning, effectively reducing trainable parameters without increasing inference latency. Due to its robust performance, LoRA and its variants have been widely used to adapt LLMs for specific tasks (Yu et al., 2024; Xu et al., 2023; Biderman et al., 2024; Chen et al., 2024; Meng et al., 2024; yang Liu et al., 2024). Benedek & Wolf (2024) and Zhang et al. (2023) show that the number of ranks required for each parameter matrix across the model's layers is not uniform. Therefore, they propose dynamically assigning ranks based on the importance of parameters during training. These rank-allocating approaches typically involve real-time estimation of parameter importance and pruning during the training process. In contrast, RaSA uses a shared rank pool combined with layer-specific weighting, eliminating the need for complex importance estimation or pruning. Biderman et al. (2024) conduct a comprehensive empirical study on LoRA, and reveal that while LoRA still lags behind FFT, it exhibits less catastrophic forgetting. We show that our proposed RaSA forgets even less than LoRA, and learns more and faster.

**Parameter Redundancy of LoRA** Although LoRA has significantly reduced the number of trainable parameters, recent research suggest that it is possible to further minimize these parameters without compromising performance. Kopiczko et al. (2024) achieve a 99% reduction in LoRA parameters by fully sharing a pair of low-rank, frozen random matrices across all layers, adjusted with learnable scaling vectors. Koohpayegani et al. (2024) propose learning linear combinations of a set of random matrix bases, while Li et al. (2024) push this further by replacing the matrix bases with a vector bank. Song et al. (2024) and Renduchintala et al. (2024) explore the effects of different sharing and selective fine-tuning strategies. By sharing parameter spaces, Brüel-Gabrielsson et al. (2024) compress 1,000 LoRAs trained from different task, enabling more efficient serving. These findings collectively suggest that LoRA's parameter has not been fully utilized and that different LoRAs exhibit similarities across layers, modules, and even different tasks. Rather than focusing on extreme parameter reduction, this work aims to maintain the same parameter count while exploring how inter-layer sharing can enhance parameter utilization. We theoretically and empirically demonstrate that sharing ranks across layers leads to lower reconstruction error and thus better expressive capacity.

## 6 CONCLUSION

In this study, we introduced RaSA, a novel extension to LoRA through an innovative partial rank sharing across layers. RaSA maintains the parameter efficiency and seamless integration into existing models characteristic of LoRA while substantially increasing the model's expressiveness. Through theoretical analysis, we established RaSA's superior capability in matrix reconstruction compared to traditional LoRA, underpinning its improved performance in downstream tasks. Empirical results on complex tasks such as code generation and mathematical reasoning have demonstrated its effectiveness over LoRA in high-demand scenarios. Future research directions may explore further optimization of rank-sharing schemes and the potential of RaSA in a broader range of applications, paving the way for the development of even more powerful and efficient PEFT strategies.

## ACKNOWLEDGMENTS

This paper is supported by the General Program of National Natural Science Foundation of China (62176153), the CCF-Tencent Rhino-Bird Open Research Fund (RAGR20240107), and the Tencent AI Lab Fund (RBFR2024002). The authors gratefully acknowledge the financial support from these funding agencies.

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

## A    COORDINATE DESCENT EXPERIMENT

This section details the derivation of the coordinate descent experiment discussed in § 3.2, inspired by Brüel-Gabrielsson et al. (2024).

Given the parameters of RaSA: $\{\tilde{\boldsymbol{B}}_i, \tilde{\boldsymbol{A}}_i, \boldsymbol{B}_S, \tilde{\boldsymbol{D}}_i, \boldsymbol{A}_S\}_{i \in [L]}$, the reconstruction error of RaSA is defined as:

$$E = \sum_{i=1}^{L} \|\boldsymbol{M}_i - (\tilde{\boldsymbol{B}}_i \tilde{\boldsymbol{A}}_i + \boldsymbol{B}_S \boldsymbol{D}_i \boldsymbol{A}_S)\|_F^2. \tag{17}$$

Clearly, $\tilde{\boldsymbol{B}}_i$ and $\tilde{\boldsymbol{A}}_i$ are independent across layers. By applying the Eckart–Young–Mirsky theorem (Eckart & Young, 1936), we first compute the SVD of the residual matrix:

$$\mathrm{SVD}(\boldsymbol{M}_i - \boldsymbol{B}_S \boldsymbol{D}_i \boldsymbol{A}_S) = \boldsymbol{U}\boldsymbol{\Sigma}\boldsymbol{V}^T. \tag{18}$$

Therefore, the update rules for $\tilde{\boldsymbol{B}}_i$ and $\tilde{\boldsymbol{A}}_i$ are:

$$\begin{aligned} \tilde{\boldsymbol{B}}_i &= \boldsymbol{U}_{[:,:r-k]}\boldsymbol{\Sigma}_{[:r-k,:r-k]}^{\frac{1}{2}}, \\ \tilde{\boldsymbol{A}}_i &= \boldsymbol{\Sigma}_{[:r-k,:r-k]}^{\frac{1}{2}}\boldsymbol{V}_{[:,:r-k]}^{T}. \end{aligned} \tag{19}$$

Let the low-rank decomposition of $\boldsymbol{M}_i$ be: $\boldsymbol{M}_i = \hat{\boldsymbol{B}}_i \hat{\boldsymbol{A}}_i$, where $\hat{\boldsymbol{B}}_i \in \mathbb{R}^{b \times R}$ and $\hat{\boldsymbol{A}}_i \in \mathbb{R}^{R \times a}$. Next, we compute the following gradients:

$$\nabla_{\boldsymbol{B}_S} E = \sum_{i=1}^{L} -2\left(\hat{\boldsymbol{B}}_i \hat{\boldsymbol{A}}_i - \left(\tilde{\boldsymbol{B}}_i \tilde{\boldsymbol{A}}_i + \boldsymbol{B}_S \boldsymbol{D}_i \boldsymbol{A}_S\right)\right)\boldsymbol{A}_S^T \boldsymbol{D}_i^T, \tag{20}$$

$$\nabla_{\boldsymbol{A}_S} E = \sum_{i=1}^{L} -2\left(\hat{\boldsymbol{B}}_i \hat{\boldsymbol{A}}_i - \left(\tilde{\boldsymbol{B}}_i \tilde{\boldsymbol{A}}_i + \boldsymbol{B}_S \boldsymbol{D}_i \boldsymbol{A}_S\right)\right)\boldsymbol{B}_S \boldsymbol{D}_i, \tag{21}$$

$$\nabla_{\boldsymbol{D}_i} E = -2\boldsymbol{B}_S^T\left(\hat{\boldsymbol{B}}_i \hat{\boldsymbol{A}}_i - \left(\tilde{\boldsymbol{B}}_i \tilde{\boldsymbol{A}}_i + \boldsymbol{B}_S \boldsymbol{D}_i \boldsymbol{A}_S\right)\right)\boldsymbol{A}_S^T, \tag{22}$$

$$\nabla_{\mathrm{diag}(\boldsymbol{D}_i)} E = \mathrm{diag}(\nabla_{\boldsymbol{D}_i} E). \tag{23}$$

By setting these gradients to zeros, we obtain the following update rules:

$$\boldsymbol{B}_S = \left(\sum_{i=1}^{L} \begin{bmatrix} \hat{\boldsymbol{B}}_i & -\tilde{\boldsymbol{B}}_i \end{bmatrix} \begin{bmatrix} \hat{\boldsymbol{A}}_i \\ \tilde{\boldsymbol{A}}_i \end{bmatrix} \boldsymbol{A}_S^T \boldsymbol{D}_i^T\right)\left(\sum_{i=1}^{L} \boldsymbol{D}_i \boldsymbol{A}_S \boldsymbol{A}_S^T \boldsymbol{D}_i^T\right)^{-1}, \tag{24}$$

$$\boldsymbol{A}_S = \left[\left(\sum_{i=1}^{L}\left(\begin{bmatrix} \hat{\boldsymbol{B}}_i & -\tilde{\boldsymbol{B}}_i \end{bmatrix}\begin{bmatrix} \hat{\boldsymbol{A}}_i \\ \tilde{\boldsymbol{A}}_i \end{bmatrix}\right)^T \boldsymbol{B}_S \boldsymbol{D}_i\right)\left(\sum_{i=1}^{L} \boldsymbol{D}_i^T \boldsymbol{B}_S^T \boldsymbol{B}_S \boldsymbol{D}_i\right)^{-1}\right]^T, \tag{25}$$

$$\mathrm{diag}(\boldsymbol{D}_i) = \left(\boldsymbol{B}_S^T \boldsymbol{B}_S \circ \boldsymbol{A}_S \boldsymbol{A}_S^T\right)^{-1}\left(\boldsymbol{B}_S^T \begin{bmatrix} \hat{\boldsymbol{B}}_i & -\tilde{\boldsymbol{B}}_i \end{bmatrix} \circ \boldsymbol{A}_S \begin{bmatrix} \hat{\boldsymbol{A}}_i \\ \tilde{\boldsymbol{A}}_i \end{bmatrix}^T\right)\mathbf{1}. \tag{26}$$

In coordinate descent, we iteratively apply Equations (19) and (24) to (26) until convergence.

## B  Prompt Templates

```
Below is an instruction that describes a task. Write a response that
appropriately completes the request.

### Instruction:
{QUESTION}

### Response:
Let's think step by step.
```

Figure 8: Evaluation prompt for mathematical reasoning.

```
Below is an instruction that describes a task. Write a response that
appropriately completes the request.

### Instruction:
{QUESTION}

### Response:
{IMPORT SECTION}

{FUNCTION SIGNATURE}
{DOCSTRING}
```

Figure 9: Evaluation prompt for code generation.

## C  Training and Data Details

**Training**  We mostly aligned our training configurations with the optimal configurations from Biderman et al. (2024). For LoRA, we used the decoupled LionW optimizer with a batch size of 192, training for 8 epochs with a learning rate of 5e-4. A cosine learning rate scheduler was applied, with the first 10% of training steps used for warmup, and weight decay set to zero. Training and evaluation were conducted using bfloat16 precision. While Biderman et al. (2024) set $\alpha = 32$ for both math and code tasks, we reduced $\alpha$ to 8 for the math task due to convergence issues observed with the Mistral model when using $\alpha = 32$. RaSA training fully inherits all hyper-parameters from LoRA training. For MoRA, we used a learning rate of 3e-4, as reported in the original work (Jiang et al., 2024). For VeRA, following the original paper, we set the learning rate to 10 times that of LoRA, resulting in 5e-3 (Kopiczko et al., 2024). All experiments for the 7-8B models were conducted on 1 node $\times$ 8 $\times$ A100-40G GPUs. For the 70B and MoE models, we used 8 nodes.

**Data**  During training, we grouped data by length, which significantly accelerated the training process. All math training data ends with "The answer is: {ANSWER}", helping answer extraction during evaluation.

