# OpenReview forum: "RaSA: Rank-Sharing Low-Rank Adaptation"
_ICLR.cc/2025/Conference — ICLR 2025 Poster_

### Official Review · Reviewer_tu4z · 2024-10-16

**Soundness:** 4
**Presentation:** 4
**Contribution:** 4
**Rating:** 8
**Confidence:** 4

**Summary:**

The paper presents a method to improve the expressivity of LoRA, while keeping the same number of parameters. This is by sharing a portion of each learnable adaptor across all layers.

**Strengths:**

* The authors tackle an important and meaningful topic that would be of interest to the community. PEFT is gaining more and more attention as the size of language models continues to grow.
* The description of the proposed method is clear and concise. Writing is easy to understand and follow.
* The method is simple, effective, and easy to implement.

**Weaknesses:**

* It seems that unlike vanilla LoRA, that acts independently on each layer, in RaSA there is an underlying assumption that all shared layers have the same dimension. If so, it is worth mentioning it explicitly in the paper.
* There are LoRA extensions that allow allocating a different rank for each layer, according to its importance (for example PRILoRA and others). It seems that in RaSA the rank should be similar across layers. If so, it is worth mentioning this fact and citing the relevant papers that do allow different rank allocation.
* There are no comparisons in Table 1 to other LoRA extensions (AdaLoRA, PRILoRA), which may be of importance to the reader. Furthermore, it is not clear why the VeRA method is compared against, when its number of parameters is x13 less.

**Questions:**

* In Figure 1b, the name of matrices A,B are in the square, and the dimension is outside. However, in matrix D the name is outside and there are no shapes. Maybe you want to consider putting the name D_i inside, and adding shape, for readability.
* In line 208, it says: ‘actual weight updates from model fine-tuning’. Do you mean ‘full fine-tuning’ (FFT)?
* In Table 1, why do you show both LAST and BEST? Why does BEST not suffice?

---

> ### Author Response · Authors · 2024-11-20
> **Response to Reviewer tu4z**
>
> Thank you for recognizing the significance, presentation, and methodology of our work!
>
>
>
> > **W1** ..., in RaSA there is an underlying assumption that all shared layers have the same dimension. If so, it is worth mentioning it explicitly in the paper.
>
> Thanks for pointing this out and we will add this assumption explicitly in the final version to make it clearly. In addition, we would like to emphasize that this assumption holds for the vast majority of widely used models (Llama, Mistral, Gemma, Qwen, ...).
>
>
>
> > **W2** There are LoRA extensions that allow allocating a different rank for each layer, according to its importance (for example PRILoRA and others). It seems that in RaSA the rank should be similar across layers. If so, it is worth mentioning this fact and citing the relevant papers that do allow different rank allocation.
>
> Thank you for pointing this out. RaSA does indeed assign a consistent rank across layers, which we will explicitly clarify in the final version of the paper. We will also include a more comprehensive discussion of rank-allocating methods such as PRILoRA and other relevant extensions, which will provide valuable context for readers.
>
>
>
> > **W3** There are no comparisons in Table 1 to other LoRA extensions (AdaLoRA, PRILoRA), which may be of importance to the reader. Furthermore, it is not clear why the VeRA method is compared against, when its number of parameters is x13 less.
>
> * We agree that including comparisons with methods like AdaLoRA and PRILoRA would provide additional context and be of interest to the reader. Please see **General Response** for details.
> * VeRA, to out knowledge, is the first work to explore the concept of sharing rank across layers. We believe this makes it an interesting and relevant point of comparison for readers.

---

> > ### Comment · Reviewer_tu4z · 2024-11-21
> >
> > I understand your point about VeRA. Perhaps you could include that explanation alongside the table to clarify your motivation for adding it.
> >
> > Regarding the comparison you made with Adalora and others, could you update the PDF so we can review it?

---

> > > ### Author Response · Authors · 2024-11-21
> > >
> > > Sure! We have updated the PDF; the additional results are in Table 3, Appendix D.

---

### Official Review · Reviewer_bzbg · 2024-10-31

**Soundness:** 4
**Presentation:** 4
**Contribution:** 4
**Rating:** 8
**Confidence:** 4

**Summary:**

Low-rank adaptation (LoRA) is limited in expression ability due to low rank constraints. This paper propose Rank-Sharing Low-Rank Adaption (RaSA) to enhance the expressive capacity of LoRA by leveraging partial rank sharing across layers. Specifically, RaSA shares $k$ ranks across all layers and assign $r-k$ for layer-specific parameters. After that, RaSA greatly increase the effective rank of the parameter update by $(L-1)\times k$ without introducing additional parameters. The experimental results on multiple datasets with different model architectures validate the effectiveness of the proposed approach.

**Strengths:**

1. The problem studied in this paper is valuable.
2. The proposed RaSA approach is interesting and effective.
3. The paper is well organization.
4. The theoretical and empirical analysis are insightful.
5. The experiment is sufficient, including the study of Reconstruction error, the study of $k$, the performance comparison of parameters and time, the study of forgetable.

**Weaknesses:**

A great paper with almost no weaknesses. A small suggestion, the distinction between the different methods in Figures 4, 5, and 7 is only based on different color (red and blue), as printing on paper cannot distinguish them.

**Questions:**

I am curious why the performance is best when $k$ is around $r/8$ (line 249-250), and if possible, is there any theoretical explanation?

---

> ### Author Response · Authors · 2024-11-20
> **Response to Reviewer bzbg**
>
> Thanks for the high assessment of this work!
>
> > **W1** A great paper with almost no weaknesses. A small suggestion, the distinction between the different methods in Figures 4, 5, and 7 is only based on different color (red and blue), as printing on paper cannot distinguish them.
>
> Thank you for this helpful suggestion! We will revise the figures in the final version to ensure better clarity.
>
>
>
> > **Q1** I am curious why the performance is best when $k$ is around $r/8$ (line 249-250), and if possible, is there any theoretical explanation?
>
> We have tried to explain this theoretically, but it is a challenging non-convex optimization problem. Therefore, so far, this remains empirical. It is worth noting that the best $k$ may also be related to the number of layers and the dimension of the model. We will clarify this in the final version. Thanks for bringing this up.

---

> > ### Comment · Reviewer_bzbg · 2024-11-26
> >
> > I maintain the original score.

---

### Official Review · Reviewer_SwqF · 2024-11-01

**Soundness:** 3
**Presentation:** 3
**Contribution:** 3
**Rating:** 6
**Confidence:** 3

**Summary:**

This paper highlights that the low-rank constraint limits the expressive capacity of LoRA, and that LoRA’s parameters are not fully utilized. To address this, the authors propose RaSA, an approach enabling partial rank sharing across layers with minimal additional parameters. Theoretical and empirical analyses demonstrate that RaSA achieves a minimum reconstruction error bounded by LoRA, and can be easily optimized. Experiments show that RaSA achieves great performance in complex math and code tasks.

**Strengths:**

- RaSA can efficiently increase the rank of the original LoRA, and compared to directly expanding the rank, RaSA achieves this with almost no increase in parameter count.
- The authors validate the better expressive capacity of RaSA through detailed reconstruction error analysis, empirical analysis shows that RaSA can achieve much lower reconstruction error and reveal the relationship between rank and the hyperparameter k.
- Experiments on code and math tasks show that RaSA achieves much better performance with only a 10% increase in time, highlighting RaSA's promising advantages.

**Weaknesses:**

- As written in Lines 30-31, LoRA still lags behind full fine-tuning, particularly on complex tasks, however, the authors only compare RaSA with existing peft methods, not including full fine-tuning.

**Questions:**

- In line 103, the authors claim that introducing a trainable diagonal matrix enables layer-specific weighting, actually, I am confused about this, maybe it could be explained specifically.

---

> ### Author Response · Authors · 2024-11-20
> **Response to Reviewer SwqF**
>
> Thank you for acknowledging the contributions of our work!
>
>
>
> > **W1** ... the authors only compare RaSA with existing peft methods, not including full fine-tuning.
>
> Thanks for the suggestion. We agree that including full fine-tuning (FFT) would enhance the completeness of our work. Please refer to **General Response** for details.
>
>
>
> > **Q1** In line 103, the authors claim that introducing a trainable diagonal matrix enables layer-specific weighting, actually, I am confused about this, maybe it could be explained specifically.
>
> Thanks for pointing this out!
>
> * Let's first explain why we call it weighting. Consider the matrices $\boldsymbol{B}=\begin{bmatrix}\boldsymbol{b}_1, \boldsymbol{b}_2, \cdots, \boldsymbol{b}_r\end{bmatrix}$, $\boldsymbol{A}=\begin{bmatrix}\boldsymbol{a}_1^T, \boldsymbol{a}_2^T, \cdots, \boldsymbol{a}_r^T\end{bmatrix}^T$, and a diagonal matrix $\boldsymbol{D}=\mathrm{diag}(d_1, d_2, \cdots, d_r)$, the product of these matrices can be expressed as:
>
>   $$\boldsymbol{B}\boldsymbol{D}\boldsymbol{A}=\sum_{i=1}^{r}d_i\boldsymbol{b}_i\boldsymbol{a}_i^T$$.
>
>   Here, each scalar $d_i$ acts as the weight of the component $\boldsymbol{b}_i\boldsymbol{a}_i^T$. This is why we call such a diagonal matrix $\boldsymbol{D}$ as a weighting.
>
> * By "trainable," we mean it is not fixed or constant.
>
> * By "layer-specific," we mean it it unique to each layer (not shared across layers).

---

> > ### Comment · Reviewer_SwqF · 2024-11-27
> >
> > Thanks for the authors' response, and I will maintain my score.

---

### Official Review · Reviewer_7i6n · 2024-11-03

**Soundness:** 2
**Presentation:** 2
**Contribution:** 3
**Rating:** 6
**Confidence:** 3

**Summary:**

The paper introduces an enhancement to existing fine-tuning methods for large language models (LLMs). It builds on Low-rank adaptation (LoRA), commonly used for efficient fine-tuning, but limited by its low-rank constraint in handling complex tasks like code generation and mathematical reasoning. To address this, the authors propose Rank-Sharing Low-Rank Adaptation (RaSA), a novel method that increases expressive capacity by sharing ranks across layers, allowing for more flexibility without adding parameter overhead. RaSA thus extends LoRA’s efficiency, achieving substantial performance gains in rigorous tasks. Code and data resources accompany the work.

**Strengths:**

1. Technical Soundness: RaSA is a technically robust extension of LoRA, overcoming its expressive limitations through partial rank sharing. Theoretical and empirical results confirm its lower reconstruction error and improved task performance.

2. Paper Presentation: The paper is clearly organized and easy to follow, with well-explained motivations, methods, and contributions, making RaSA's innovations accessible and engaging.

3. Theoretical Analysis: Theoretical bounds and empirical evidence validate RaSA’s improved capacity over LoRA, providing a solid foundation for its enhanced performance in high-rank tasks.

**Weaknesses:**

1. Lack of Baseline Comparisons:
The paper lacks a comprehensive comparison with other LoRA variations, such as EM-LoRA, AdaLoRA, and Orthonormal LoRA. Without these baselines, it’s difficult to fully assess the advantages of RaSA, as the performance improvements over the original LoRA may not necessarily hold when compared with these alternative methods.

2. Efficiency Concerns:
The proposed method is considerably less efficient than the original LoRA, with a noticeable increase in computational time (at least 1.4 hours). Despite this added overhead, the performance gains are minimal, with PASS@1 improvements generally below 2%. This limited payoff questions the practical viability of RaSA, especially for real-world applications where computational efficiency is crucial.

**Questions:**

1. Could the efficiency of the proposed method be further optimized to reduce computational time?

2. The performance improvement appears to be notably greater on Mistral-0.3-7B compared to Llama-3.1-8B. Do you have any insights into the reasons behind this difference?

---

> ### Author Response · Authors · 2024-11-20
> **Response to Reviewer 7i6n**
>
> Thank you for the positive feedback and helpful suggestions!
>
>
>
> > **W1** Lack of baseline comparisions
>
> Please see **General Response**.
>
> Additionally, we note that EM-LoRA is still an [anonymous submission](https://openreview.net/forum?id=MpXSpER30w) to acl rolling review without open-sourced code. Therefore, we have not added EM-LoRA as a baseline and would be willing to add a disscusion about it in our final version.
>
>
>
> > **W2 & Q1** Efficiency Concerns
>
> * **Training Overhead:** We consider the extra computational time during training to be acceptable. The following table shows the time increments relative to LoRA, with RaSA incurring ~15% additional time. Notably, when compared to another high-rank method, MoRA, this 15% overhead is modest. Reviewer SwqF also agrees with this in Strengths (*... with only a 10% increase in time, highlighting RaSA's promising advantages*).
>
>   | r    | Method   | Llama-3.1-8B | Mistral-0.3-7B |
>   | ---- | -------- | ------------ | -------------- |
>   | 8    | MoRA     | 25.0%        | 25.2%          |
>   |      | **RaSA** | **16.7%**    | **13.1%**      |
>   | 16   | MoRA     | 29.6%        | 30.8%          |
>   |      | **RaSA** | **14.3%**    | **13.1%**      |
>   | 32   | MoRA     | 24.0%        | 29.6%          |
>   |      | **RaSA** | **15.0%**    | **15.7%**      |
>
>   Furthermore, we argue that **parameter efficiency** and **inference efficiency** are more critical metrics than training time when evaluating the efficiency of a PEFT method, as training is typically a one-time cost. RaSA fully inherits the advantages of LoRA in these two aspects (*line 47-48: RaSA retains the core advantages of LoRA – keeping the same parameter overhead and allowing for*
>   *easy merging back into the model*).
>
> * **Performance Improvement:** In Table 1, RaSA outperforms LoRA in PASS@1 by 5.4% in average, not 2%.
>
>
>
> > **Q2** The performance improvement appears to be notably greater on Mistral-0.3-7B compared to Llama-3.1-8B. Do you have any insights into the reasons behind this difference?
>
> That's a good question. We speculate that RaSA is more effective on tasks that are challenging relative to the model’s capacity. For the math and code tasks in our evaluation, these challenges are more pronounced for Mistral-0.3-7B than for Llama-3.1-8B (confirmed in Meta's [blog](https://ai.meta.com/blog/meta-llama-3-1/) of llama-3.1), which explains the larger performance gains.

---

> > ### Comment · Reviewer_7i6n · 2024-11-27
> >
> > Thanks for the author's response. I will keep my score as it is.

---

### Author Response · Authors · 2024-11-20
**General Response**

We appreciate the reviewers for their thorough review and valuable feedback.

Based on the reviewers' suggestions, we have decided to add FFT, OLoRA [1], AdaLoRA [2] and PriLoRA [3] for comparison. Due to time constraints, we present partial results below (fine-tuning Mistral model on math task). The full results will be included in the final version of the paper.

| # Trainable Params. | # Extra params. | Method  | BEST     | LAST     |
| ------------------- | --------------- | ------- | -------- | -------- |
| -----               | -----           | FFT     | 28.1     | 26.6     |
| **r=8**             | -----           | -----   | -----    | -----    |
| 21.0M               | 00.0M           | LoRA    | 20.1     | 19.2     |
| 21.0M               | 00.0M           | MoRA    | 21.4     | 21.2     |
| 21.0M               | 00.0M           | OLoRA   | 22.5     | 22.5     |
| 31.5M               | 63.0M           | AdaLoRA | 22.5     | 21.6     |
| 21.3M               | 10.7M           | PriLoRA | 22.3     | 22.3     |
| 21.0M               | 00.0M           | RaSA    | **24.3** | **23.8** |
| **r=16**            | -----           | -----   | -----    | -----    |
| 41.9M               | 00.0M           | LoRA    | 20.9     | 19.5     |
| 41.9M               | 00.0M           | MoRA    | 20.5     | 19.4     |
| 41.9M               | 00.0M           | OLoRA   | 22.5     | 22.2     |
| 62.9M               | 126.8M          | AdaLoRA | 23.5     | 23.2     |
| 42.6M               | 21.3M           | PriLoRA | 22.7     | 21.6     |
| 42.0M               | 00.0M           | RaSA    | **25.9** | **25.1** |

*Note: for PriLoRA, we set $r_s=0.5r$ and $r_f=1.5r$* as done in [3].

* **FFT indeed performs significantly better than LoRA**, consistent with the findings in [4]. Our method significantly bridges the gap between LoRA and FFT, achieving the best performance among PEFT methods.
* **Rank-allocating methods such as AdaLoRA and PriLoRA introduce additional parameters**. AdaLoRA starts with a 1.5x parameter budget, reducing to 1x at the end of training, with peak trainable parameters at 1.5x LoRA. It also introduces sensitivity and uncertainty matrices, adding 2x the extra parameter count. Similarly, PriLoRA includes a mask matrix, contributing 0.5x the extra parameters.

**References**

[1] OLoRA: Orthonormal Low-Rank Adaptation of Large Language Models, Büyükakyüz et al., arXiv:2406.01775

[2] AdaLoRA: Adaptive Budget Allocation for Parameter-Efficient Fine-Tuning, Zhang et al., ICLR 2023

[3] PRILoRA: Pruned and Rank-Increasing Low-Rank Adaptation, Benedek et al., EACL 2024

[4] LoRA Learns Less and Forgets Less, Biderman et al., TMLR 2024

---

> ### Comment · Reviewer_tu4z · 2024-11-21
>
> Thank you. Can you kindly update the PDF so we can see it in context?

---

> ### Author Response · Authors · 2024-11-21
> **PDF updated**
>
> Dear Reviewers,
>
> We have updated the PDF to reflect the corresponding revisions (highlighted in light blue). The additional baselines are placed in Table 3 of Appendix D (as they are still being completed).
>
> Thank you!

---

### Comment · Area_Chair_RAUU · 2024-11-22
**Interactive Discussions**

Dear Reviewers,

Thank you for your efforts in reviewing this paper. We highly encourage you to participate in interactive discussions with the authors before November 26, fostering a more dynamic exchange of ideas rather than a one-sided rebuttal.

Please feel free to share your thoughts and engage with the authors at your earliest convenience.

Thank you for your collaboration.

Best regards,
ICLR 2025 Area Chair

---

### Meta-Review · Area_Chair_RAUU · 2024-12-20

**Metareview:**

This submission highlights that the low-rank constraint in LoRA limits its expressive capacity, resulting in underutilization of LoRA's parameters. To address this issue, the authors propose RaSA, a method that enables partial rank sharing across layers with minimal additional parameters. Both theoretical and empirical analyses demonstrate that RaSA achieves a reconstruction error bounded by LoRA, while remaining easy to optimize. Experimental results show that RaSA delivers strong performance on complex math and code tasks.

After the rebuttal, all reviews recommended acceptance, including two clear acceptances. The area chair concurs, recognizing that this submission makes a significant contribution to the PEFT field, and recommends acceptance.

**Additional Comments On Reviewer Discussion:**

Almost all concerns were addressed during the rebuttal phase, and the authors are encouraged to incorporate these clarifications into the final version.

---

### Decision · Program_Chairs · 2025-01-22

Accept (Poster)